# The Effect of Educational Expectations on Children’s Cognition and Depression

**DOI:** 10.3390/ijerph192114070

**Published:** 2022-10-28

**Authors:** Meimei Liu, Tao Zhang, Ning Tang, Feng Zhou, Yong Tian

**Affiliations:** 1School of Food Science, Nanjing Xiaozhuang University, Nanjing 211171, China; 2Wuxi Big Bridge Academy, Wuxi 214000, China; 3School of Public Health, Southeast University, Nanjing 210009, China

**Keywords:** parents’ educational expectations, children’s educational expectations, differences in educational expectations, cognition, depression

## Abstract

Cognitive and psychological conditions in childhood will have an important impact on adult life. There is relatively little literature on the impact of educational expectations on children’s cognition and psychological health from the perspective of urban and rural differences. Based on the cohort data of the CFPS from 2012 and 2016, this study screened a total of 994 children aged 10–15 to study the effects of parents’ educational expectations and children’s educational expectations on children’s cognition and depression. The results show that both parents’ educational expectations and children’s educational expectations have a positive impact on children’s cognition. Parents’ educational expectations and children’s educational expectations have negative effects on children’s depression. When parents’ educational expectations are greater than their children’s educational expectations, educational expectations have a negative impact on children’s cognition and a positive impact on children’s depression. In both urban and rural samples, parents’ educational expectations and children’s educational expectations have a positive impact on children’s cognition and a negative impact on children’s depression. However, the impact of educational expectations on children’s cognition and depression was greater in rural areas than in urban areas. When parents’ educational expectations are greater than their children’s educational expectations, educational expectations in urban areas have no effect on children’s cognition.

## 1. Introduction

Individual cognitive ability plays an important role in various achievements in the later life-course. Studies have shown that the formation of an individual’s early cognitive ability is the basis for his future achievement, mainly because early experiences can have a long-term and important impact on a person’s future development [1,2]. Cognitive ability plays an important role in children’s development, not only affecting their academic performance but also their future occupational status and potential for development to the upper echelons of society [3,4,5]. Therefore, the study of cognitive development in childhood is of great significance.

In the research literature on the influencing factors of cognitive ability formation, most studies focus on individual environmental factors, especially individual family environmental factors [6,7]. Family environment and parental investment are important input factors for children’s cognitive ability [8]. Wealthy families invest more in childhood than poor families, creating persistent cognitive differences [9]. Family background directly affects children’s academic performance and cognitive ability and also indirectly affects academic performance and cognitive ability through the impact on children’s efforts [10].

Educational investment is directly related to cognitive ability level [11]. Previous studies have shown that parental involvement contributes to the formation of a closed “home–school–society” relationship, which can help improve children’s cognitive abilities [12,13]. In family-based parental participation, parents spend more time with their children reading books, playing games, and communicating with their children, which helps to improve children’s cognitive ability [14,15]. There are also studies that suggest that parents’ educational expectations have a positive impact on the cognitive ability of Chinese-American students [16]. Research on China also mainly focuses on the group of left-behind children, and the conclusions of the research are controversial [17].

Parents’ educational expectations and children’s educational expectations will have an important impact on children’s development. The higher parents’ expectations for their children’s education, the more likely they are to participate in extracurricular education [18]. In addition, the improvement of parents’ educational expectations can also reduce the level of academic burnout in children [19]. An increase in self-education expectations can significantly improve the chances of getting a college education [20]. In addition, the expectation of self-education has a positive impact on the improvement of children’s word ability and mathematical ability [21].

In addition, the psychological status of children is also one of the most important factors affecting the development of children. From the perspective of children’s psychological development, the main influencing factors in the family are: family socioeconomic status, parenting style, the parent–child relationship, parents’ character, family structure, living environment, parents’ educational expectations, etc. [22,23,24,25]. Parental expectations reinforce parents’ educational behavior and prompt them to care more about their children, but the positive effects of educational expectations must be based on moderation. Parents’ good educational expectations will be transformed into children’s achievement motivation to enhance their self-confidence, while excessive expectations will cause children’s psychological pressure, and in severe cases, will lead to children’s low self-esteem and dampen their self-confidence [26]. If parents’ educational expectations are too high and exceed the child’s ability, the anxiety of the child will increase, which will have a negative impact on the child’s learning; but too low educational expectations may waste the child’s talent and fail to fully tap its potential. It will make the child’s motivation to learn achievement decline and make it fail to develop fully, which will be detrimental to the child’s growth [27]. In addition, studies have shown that when parents’ educational expectations are higher than their children’s educational expectations, children’s psychological conditions will be worse; however, the study was limited to left-behind children in rural China [28].

Cognitive ability and depression are both important components of a child’s health and an important factor influencing a child’s future development. Furthermore, cognitive and mental health also interact with each other. The family environment and parental involvement influence children’s cognition and depression. Current research has demonstrated that parents’ educational expectations have an impact on children’s cognition and depression. However, these studies still have some shortcomings, mainly in the following aspects: First, there are relatively few studies on China, and they mainly focus on rural left-behind children. Secondly, there is no research on the impact of self-education expectations and educational expectation gaps on children’s cognition. Therefore, the marginal contribution of this study is mainly reflected in the following aspects: (1) The effects of parental expectations and self-expectations on children’s cognition and depression were evaluated in a full urban and rural sample. (2) This study assessed the impact of the gap between parents’ educational expectations and their children’s educational expectations on children’s cognition and depression. Based on this, we propose the following research hypotheses:

**Hypothesis** **1** **(H1).***Parents’ educational expectations and children’s educational expectations have a positive role in promoting children’s cognition*.

**Hypothesis** **2** **(H2).**
*The improvement of parents’ educational expectations and children’s educational expectations can effectively reduce the probability of children’s depression.*


**Hypothesis** **3** **(H3).**
*When parents’ educational expectations exceed their children’s educational expectations, the improvement of educational expectations is not conducive to the improvement of children’s cognition but they will increase the risk of depression in children.*


**Hypothesis** **4** **(H4).**
*There are urban–rural differences in the effects of parents’ educational expectations, children’s educational expectations, and the educational expectations’ gap on children’s cognition and depression.*


## 2. Materials and Methods

### 2.1. Data

This study is based on data from two waves of the China Family Panel Studies (CFPS) from 2012 and 2016 (Figure 1). CFPS is a nationwide survey data covering 25 provinces, municipalities, and autonomous regions in China. It aims to reflect the changes in China’s society, economy, population, education, and health by tracking and collecting data at three levels: individual, family, and community, which provides a data foundation for academic research and public policy analysis. CFPS conducted a baseline survey in 2010 and conducts follow-up surveys every 2 years. CFPS uses face-to-face interviews, telephone interviews, and computer-assisted technology to obtain information about respondents’ personal, family, and community perspectives. Due to differences in depression measurement and cognitive measurement methods of CFPS in different years, in order to maintain the consistency of measurement standards, this study selected the data from 2012 and 2016 that were consistent in cognitive and depression measurement standards. At the same time, this study mainly focused on children who were able to autonomously answer questions about cognition and depression. Therefore, this study selected children aged 10–15 as the research sample. Ethical approval was granted by the institutional review board (IRB00001052-14010). All participants provided written informed consent [29]. 

### 2.2. Mesures

#### 2.2.1. Cognition (Dependent Variable)

This study divided cognitive measures into two parts: word recall ability and number sequence ability. Among them, word recall ability was divided into two parts: timely recall and delayed recall. After hearing 10 Chinese words, respondents were asked to recall and repeat as many words as possible. Every time a word was answered correctly, it was counted as 1 point, and the timely recall score ranged from 0 to 10. After 2 min, respondents were again asked to recall as many words as possible, known as delayed memory. Each time a word was answered correctly, 1 point was counted, and the score for delayed recall ranged from 0 to 10. The word recall score was the average of the timely recall and delayed recall scores, and its value ranged from 0 to 10. In the sequence test, a line of numbers was displayed on the respondent’s computer screen, one of which was blank, and the respondent was asked to fill in the appropriate number in the blank. There were 15 questions in each group. Respondents obtained 1 point for each correct answer. The score range of the sequence test was 0–15. The total cognitive ability score was the sum of the word recall score and the sequence score, which ranged from 0 to 25.

#### 2.2.2. Depression (Dependent Variable)

This study combined the CES-D Depression Scale with CFPS data to assess depression in children. In the CFPS questionnaire, respondents were asked 20 questions related to their mental state. There were four options for each question, namely 1. Almost never (less than a day), 2. Sometimes (1–2 days), 3. Often (3–4 days), and 4. Most of the time (5–7 days). Respondents who chose options 1–4 would be given 0–3 points, respectively. The depression score was the sum of the scores from 20 to the question, and the value range was 0–60. The higher the score, the more obvious the depression tendency.

#### 2.2.3. Parents’ Educational Expectations (Independent Variable)

Parents’ educational expectations were a multi-categorical variable. In the CFPS questionnaire, parents were asked what level of education they want their children to have. There were 8 options from not having to school to a Ph.D. In this study, the options were combined into 5 levels, namely 0. No need to go to school, 1. Elementary or junior high school, 2. High school, 3. College or undergraduate, and 4. Master’s or Ph.D.

#### 2.2.4. Self-Education Expectations (Independent Variable)

Self-education expectations were a multi-categorical variable. In the CFPS questionnaire, respondents were asked what level of education they expect. In this study, the answers were divided into 5 levels, namely 0. No need to go to school, 1. Elementary or junior high school, 2. High school, 3. College or undergraduate, and 4. Master’s or Ph.D.

#### 2.2.5. Difference in Educational Expectations (Independent Variable)

In this study, the educational expectation difference was calculated by subtracting the educational expectation of the parents from the educational expectation of the children. If the educational expectation difference was greater than or equal to 0, the value was 1; otherwise, the value was 0 [28].

#### 2.2.6. Covariates

A range of covariates was incorporated into the model as follows: age (continuous variable, and the value range was 10–15), gender (male or female), self-assessed health (extremely healthy, very healthy, relatively healthy, generally, or unhealthy), per capita household income (minimum 25%, middle and lower 25%, middle and upper 25%, and up to 25%), number of times they see their parents per week (continuous variable), tutoring class (no or yes), and arguments with parents (continuous variable).

### 2.3. Statistical Analysis

This study used stata16 software (Stata Corp, College Station, TX, USA) for data analysis. This study firstly performed descriptive statistical analysis on the sample participating in the model. Then, a linear panel fixed-effects model was used to evaluate the effects of parents’ educational expectations, children’s educational expectations, and the difference in educational expectations on children’s cognition and depression. Finally, urban–rural differences in this effect were analyzed using a panel fixed-effects model.

## 3. Results

### 3.1. Descriptive Statistical Analysis

Table 1 shows the results of the descriptive statistical analysis. From the results, a total of 994 children aged 10–15 were included in the analysis sample. Among them, urban areas accounted for 357, and rural areas accounted for 637. In the full sample, men accounted for 51.61%, and women accounted for 48.39%. The mean age of the sample was 12.33 years (SD = 1.76). The average cognitive ability score for the full sample was 15.91 (SD = 7.84). Among them, the average score of cognitive ability in urban areas was 15.99 (SD = 8.13), and the average score of cognitive ability in rural areas was 15.87 (SD = 7.68). The mean depression score of the whole sample was 8.00 (SD = 6.32). Among them, the average depression score of the urban sample was 7.65 (SD = 6.10), and the average depression score of the rural sample was 8.19 (SD = 6.44). In terms of parents’ educational expectations, the proportion of children who wanted their children not to go to school was 0, the proportion of children who wanted their children to go to primary school or junior high school was 7.04%, the proportion of children who wanted their children to go to high school at most was 37.22%, the proportion of children who wanted their children to go to college or university at most was 51.01%, and the proportion of children who wanted their children to obtain a Master’s or Ph.D. was 4.73%. In terms of personal self-expectation, the proportion of children who wanted their children not to go to school was 0, the proportion of children who wanted their children to go to primary school or junior high school was 9.39%, the proportion of children who wanted their children to go to high school at most was 43.84%, the proportion of children who wanted their children to go to college or undergraduate at most was 41.82%, and the proportion of children who wanted their children to obtain a Master’s or Ph.D. was 4.95%. In terms of the difference in educational expectations, the proportion of parents’ expectations greater than or equal to their children’s expectations was 92.63%. The proportion of parents’ expectations less than their children’s expectations was 7.37%.

In this study, an χ^2^ test (for the categorical variables) and a Kruskal–Wallis test (for the continuous variable) were used to examine the differences between the urban group and the rural group in each included variable. The results of the study showed that there were significant differences (*p* < 0.1) between the urban group and the rural group in terms of cognition, CESD, parents’ educational expectations, children’s educational expectations, the difference in educational expectations, gender, age, self-assessed health, per capita household income, the number of times per week to see parents, tutoring class, and arguments with parents.

### 3.2. Benchmark Regression Results

Table 2 shows the regression results of educational expectations on children’s cognition and depression. Model 1 shows the regression results of parents’ educational expectations on children’s cognition. The findings show that parents’ educational expectations had a positive effect on their children’s cognition at a 1% confidence level (β = 3.38, CI: 2.76~4.40, *p* < 0.001). The results of Model 2 show that children’s educational expectations had a positive effect on children’s cognition at the 1% confidence level (β = 2.51, CI: 1.87~3.16, *p* < 0.001). The results of Model 3 show that parents’ educational expectations had a negative inhibitory effect on children’s depression at the 1% significance level (β = −2.87, CI: −3.42~−2.32, *p* < 0.001). The results of Model 4 show that children’s educational expectations had a negative inhibitory effect on children’s depression at the 1% significance level (β = −2.15, CI: −2.68~−1.62, *p* < 0.001).

The results of Model 5 show that the differences in educational expectations had a negative inhibitory effect on children’s cognition at the 1% significance level (β = −3.31, CI: −5.28~−1.33, *p* < 0.001). The results of Model 6 show that the differences in educational expectations had a positive effect on children’s depression at the 1% significance level (β = 2.20, CI: 0.92~3.47, *p* < 0.001).

Table 3 shows the urban–rural differences in children’s cognition by educational expectations. In the urban samples, parents’ educational expectations (β = 2.89, CI: 1.81~3.98, *p* < 0.001) and children’s educational expectations (β = 2.05, CI: 0.91~3.20, *p* < 0.001) had a positive effect on children’s cognition. However, the differences in educational expectations had no effect on children’s cognition (β = −2.90, CI: −6.53~0.74, *p* = 0.12).

Parents’ educational expectations (β = −2.35, CI: −3.20~−1.50, *p* < 0.001) and children’s educational expectations (β = −1.77, CI: −2.64~−0.91, *p* < 0.001) had a negative inhibitory effect on children’s depression. The differences in educational expectations had a positive effect on children’s depression (β = 2.96, CI: 0.24~5.68, *p* < 0.001).

In the rural samples, parents’ educational expectations (β = 3.49, CI: 2.73~4.24, *p* < 0.001) and children’s educational expectations (β = 2.46, CI: 1.70~3.22, *p* < 0.001) had a positive effect on children’s cognition. The differences in educational expectations had a negative inhibitory effect on children’s cognition (β = −2.59, CI: −5.24~−0.66, *p* = 0.01).

Parents’ educational expectations (β = −3.00, CI: −3.72~−2.28, *p* < 0.001) and children’s educational expectations (β = −2.07, CI: −2.70~−1.44, *p* < 0.001) had a negative inhibitory effect on children’s depression. The differences in educational expectations had a positive effect on children’s depression (β = 1.85, CI: 0.30~3.40, *p* < 0.001).

## 4. Discussion

### 4.1. Results Discussion

To the best of our knowledge, this is the first study in mainland China to examine the impact of educational expectations on children’s cognition and depression from the perspective of urban–rural differences. This study used CFPS2012 and CFPS2016 panel data to evaluate the impact of parents’ educational expectations and children’s educational expectations on children’s cognition and depression, and it analyzed the urban–rural differences in this effect.

First, this study found that the improvement of parents’ educational expectations and children’s educational expectations helps to promote children’s cognitive improvement. Studies have shown that education is one of the most important factors affecting the development of children’s cognitive ability [1]. From the perspective of the source of cognitive ability investment, family investment is one of the decisive factors for the development of children’s cognitive ability [30,31]. Parents’ educational expectations affect their children’s academic performance and mental health. When parents’ educational expectations increase, the investment in children’s education will also increase, which in turn has an impact on children’s cognitive ability [32]. In addition, existing research has demonstrated that adolescents’ self-education expectations also have a positive impact on learning engagement and academic performance [21]. This is basically consistent with the conclusion of this study.

Second, this study found that the improvement of parents’ educational expectations and children’s educational expectations helps reduce the probability of depression. Studies have shown that increased educational expectations are conducive to improved academic performance, and improved academic performance is conducive to improved mental health [21]. However, high educational expectations have negative effects on mental health, especially for after-school tutoring to improve academic performance [33]. It can be seen that the effects of parents’ educational expectations and children’s educational expectations on children’s mental health show different effects under different conditions.

Finally, this study found that when parents’ educational expectations exceed their children’s educational expectations, their educational expectations reduce a child’s cognition and increase the probability of depression. Among Chinese teenagers aged 10–15, 40–60% of parents’ educational expectations deviate from their personal educational expectations. Most parents have high educational expectations [34]. When parents’ educational expectations are too high, it easily to leads to quarrels between parents and children, which will adversely affect the mental health of teenagers [28]. In particular, it is important to note that parents’ high expectations for their children do not necessarily lead to anxiety, but deviations in expectations can lead to anxiety [35]. In addition, existing studies have also shown that a parent–child education expectation bias has a negative impact on academic performance [36]. Family social capital can only be effectively transmitted through frequent parent–child communication and parent–child companionship when parents and children have consistently high educational expectations. When the educational expectations of parents and students are inconsistent, the effect of high educational expectations will be weakened due to the lack of social capital caused by parents’ low educational expectations or the ineffective transmission of social capital caused by students’ low educational expectations [37]. Therefore, parents’ educational expectations and children’s educational expectations need to be consistent in order to promote children’s cognitive and mental health. When parents’ educational expectations are greater than their children’s educational expectations, educational expectations will have the opposite effect on children’s cognition and depression.

### 4.2. Limitations and Research Prospects

The limitations of this study are mainly reflected in the following aspects: First, due to the data and space limitations, no further mechanistic studies were performed in this study. The influence of educational expectations on children’s cognition and depression may be through educational investment, extracurricular training, etc. This needs to be further explored in future studies. Second, although this study used panel data to evaluate the impact of educational expectations on children’s cognition and depression, there may still be endogeneity problems. Further research can address this issue through the instrumental variable approach.

## 5. Conclusions

This study used CFPS2012 and CFPS2016 panel data to evaluate the effects of parents’ educational expectations, children’s educational expectations, and the differences in educational expectations on children’s cognition and depression. The study found that parents’ educational expectations and children’s educational expectations have a positive promoting effect on children’s cognition and a negative inhibitory effect on children’s depression. However, when parents’ educational expectations exceed their children’s educational expectations, their educational expectations have a negative inhibitory effect on children’s cognition and a positive stimulating effect on children’s depression. In both urban and rural samples, parents’ educational expectations and children’s educational expectations have a positive impact on children’s cognition and a negative impact on children’s depression. However, the impact of educational expectations on children’s cognition and depression was greater in rural areas than in urban areas. When parents’ educational expectations are greater than their children’s educational expectations, educational expectations in urban areas have no effect on children’s cognition, while educational expectations in rural areas have a negative impact on children’s cognition. At this time, regardless of urban or rural areas, educational expectations have a positive impact on children’s depression.

## Figures and Tables

**Figure 1 ijerph-19-14070-f001:**
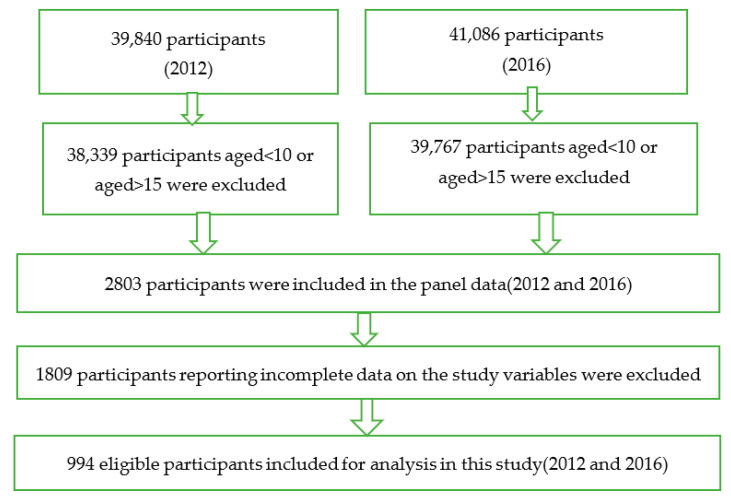
A flow chart of the study sample from 2012 to 2016.

**Table 1 ijerph-19-14070-t001:** Descriptive statistical analysis.

Characteristics	All (*n* = 994)	Urban (*n* = 357)	Rural (*n* = 637)	*p* Value ^a^
Cognition (Mean + SD)	15.91 ± 7.84	15.99 ± 8.13	15.87 ± 7.68	<0.001
CESD (Mean + SD)	8.00 ± 6.32	7.65 ± 6.10	8.19 ± 6.44	<0.001
Parents’ educational expectations				<0.001
No need to go to school	0	0	0	
Elementary or junior high school	70 (7.04%)	17 (4.76%)	53 (8.32%)	
High school	370 (37.22%)	133 (37.25%)	237 (37.21%)	
College or undergraduate	507 (51.01%)	185 (51.82%)	322 (50.55%)	
Master’s or Ph.D.	47 (4.73%)	22 (6.16%)	25 (3.92%)	
Children’s educational expectations				<0.001
No need to go to school	0	0	0	
Elementary or junior high school	93 (9.39%)	17 (4.79%)	76 (11.97%)	
High school	434 (43.84%)	155 (43.66%)	279 (43.94%)	
College or undergraduate	414 (41.82%)	162 (45.63%)	252 (39.69%)	
Master’s or Ph.D.	49 (4.95%)	21 (5.92%)	28 (4.41%)	
Difference in educational expectations				0.061
≥0	917 (92.63%)	328 (92.39%)	589 (92.76%)	
<0	73 (7.37%)	27 (7.61%)	46 (7.24%)	
Gender				<0.001
Male	513 (51.61%)	191 (53.50%)	322 (50.55%)	
Female	481 (48.39%)	166 (46.50%)	315 (49.45%)	
Age	12.33 ± 1.76	12.27 ± 1.71	12.36 ± 1.79	<0.001
Self-assessed health				0.001
Extremely healthy	301 (30.28%)	117 (32.77%)	184 (28.89%)	
Very healthy	356 (35.81%)	126 (35.29%)	230 (36.11%)	
Relatively healthy	274 (27.57%)	96 (26.89%)	178 (27.94%)	
Generally	53 (5.33%)	16 (4.48%)	37 (5.81%)	
Unhealthy	10 (1.01%)	2 (0.56%)	8 (1.26%)	
Per capita household income				<0.001
Minimum 25%	328 (33.00%)	67 (18.77%)	261 (40.97%)	
Middle and lower 25%	333 (33.50%)	115 (31.21%)	218 (34.22%)	
Middle and upper 25%	227 (22.84%)	114 (31.93%)	113 (17.74%)	
Up to 25%	106 (10.66%)	61 (17.09%)	45 (7.06%)	
Number of times per week to see parents	3.86 ± 3.06	4.93 ± 2.87	3.26 ± 3.00	<0.001
Tutoring class				<0.001
No	869 (87.42%)	279 (78.15%)	590 (92.62%)	
Yes	125 (12.58%)	78 (21.85%)	47 (7.38%)	
Arguments with parents	0.97 ± 2.93	1.18 ± 3.38	0.85 ± 2.64	<0.001

Note: SD = standard error. ^a^ *p* values were calculated from an χ^2^ test (for categorical variables) or a Kruskal–Wallis test (for the continuous variable).

**Table 2 ijerph-19-14070-t002:** (**A**) Benchmark regression results. (**B**) Benchmark regression results.

A	Cognition		Depression	
	Model 1	Model 2	Model 3	Model 4
	β (CI) *p*-Value	β (CI) *p*-Value	β (CI) *p*-Value	β (CI) *p*-Value
Parents’ educational expectations	3.38 (2.76, 4.40) ***		−2.87 (−3.42, −2.32) ***	
Children’s educational expectations		2.51 (1.87, 3.16) ***		−2.15 (−2.68, −1.62) ***
Age	0.68 (0.43, 0.93) ***	0.79 (0.54, 1.05) ***	−0.28 (−0.50, 0.06) **	−0.40 (−0.63, −0.18) ***
Gender	−0.79 (−1.70, 0.12) *	−0.65 (−1.58, 0.29)	−0.60 (−1.31, 0.12)	−0.77 (−1.51, −0.03) *
Self-assessed health				
very healthy	−0.03 (−1.17, 1.11)	−0.03 (−1.19, 1.14)	0.86 (−0.01, 1.72) *	0.85 (−0.04, 1.74)
relatively healthy	1.00 (−0.21, 2.21)	1.01 (−0.25, 2.26)	2.08 (1.11, 3.06)	2.14 (1.14, 3.13) ***
generally	−0.24 (−2.33, 1.86)	−0.67 (−2.78, 1.44)	1.98 (0.22, 3.74)	2.32 (0.62, 4.02) **
unhealthy	−5.40 (−10.89, 0.08) *	−5.28 (−10.87, 0.30) *	3.47 (−0.23, 7.16)	3.24 (−0.21, 6.69)
Per capita household income				
Middle and lower 25%	−0.74 (−1.85, 0.37)	−0.85 (−1.99, 0.28)	0.34 (−0.62, 1.31)	0.40 (−0.60, 1.40)
Middle and upper 25%	−0.56 (−1.80, 0.68)	−0.80 (−2.07, 0.48)	−0.16 (−1.15, 0.82)	−0.01 (−1.02, 0.99)
Up to 25%	1.29 (−0.35, 2.93)	0.97 (−0.71, 2.65)	−0.03 (−1.26, 1.21)	0.15 (−1.13, 1.44)
Number of times per week to see parents	−0.07 (−0.23, 0.08)	−0.09 (−0.24, 0.07)	0.22 (0.09, 0.34) ***	0.22 (0.09, 0.35) **
Tutoring class	0.40 (−0.88, 1.67)	0.46 (−0.85, 1.77)	−0.76 (−1.74, 0.22)	−0.82 (−1.84, 0.20)
Arguments with parents	0.03 (−0.12, 0.19)	0.07 (−0.12, 0.25)	0.33 (0.20, 0.46) ***	0.35 (0.20, 0.49) ***
Constant	−0.40 (−3.95, 3.16)	0.74 (−2.89, 4.38)	16.87 (13.87, 19.88) ***	16.30 (13.13, 19.46) ***
R^2^	0.65	0.61	0.31	0.37
Observations	994	990	994	990
**B**	**Cognition**		**Depression**	
	**Model 5**		**Model 6**	
	**β (CI) *p*-Value**		**β (CI) *p*-Value**	
Educational expectations gap	−3.31 (−5.28, −1.33) ***		2.20 (0.92, 3.47) ***	
Age	0.92 (0.66, 1.18) ***		−0.50 (−0.74, −0.27) ***	
Gender	−0.53 (−1.48, 0.43)		−0.86 (−1.62, −0.11) *	
Self-assessed health				
very healthy	−0.15 (−1.37, 1.07)		0.91 (−0.03, 1.84)	
relatively healthy	0.79 (−0.51, 2.08)		2.30 (1.27, 3.33) ***	
generally	−1.27 (−3.44, 0.90)		2.80 (1.01, 4.59) ***	
unhealthy	−5.44 (−10.94, 0.05) *		3.43 (−0.01, 6.86)	
Per capita household income				
Middle and lower 25%	−0.62 (−1.80, 0.55)		0.23 (−0.83, 1.29)	
Middle and upper 25%	−0.49 (−1.80, 0.81)		−0.25 (−1.31, 0.81)	
Up to 25%	1.34 (−0.49, 3.16)		−0.18 (−1.49, 1.14)	
Number of times see parents per week	−0.12 (−0.28, 0.05)		0.26 (0.12, 0.40) ***	
Tutoring class	0.85 (−0.47, 2.17)		−1.16 (−2.24, −0.07) *	
Arguments with parents	0.08 (−0.10, 0.26)		0.34 (0.20, 0.47) ***	
Constant	8.28 (4.26, 12.29) ***		10.28 (7.19, 13.37) ***	
R^2^	0.58		0.36	
Observations	990		990	

Note: Coefficients, 95% confidence intervals, and *p*-values are reported in this table. * *p* < 0.1. ** *p* < 0.05. *** *p* < 0.01. The model uses robust standard errors.

**Table 3 ijerph-19-14070-t003:** (**A**) Benchmark regression results (Urban). (**B**) Benchmark regression results (Rural).

A	Cognition			Depression		
	Model 7	Model 8	Model 9	Model 10	Model 11	Model 12
Parents’ educational expectations	2.89 *** (1.81–3.98)			−2.35 *** (−3.20, −1.50)		
Children’s educational expectations		2.05 *** (0.91, 3.20)			−1.77 *** (−2.64, −0.91)	
Educational expectations gap			−2.90 (−6.53, 0.74)			2.96 ** (0.24, 5.68)
covariates	Yes	Yes	Yes	Yes	Yes	Yes
Constant	−1.09 (−7.78, 5.60)	−2.70 (−9.64, 4.25)	5.08 (−2.33, 12.48)	14.78 (9.30, 20.26)	13.53 (7.98, 19.08)	9.08 (3.41, 14.75)
R^2^	0.70	0.66	0.69	0.10	0.21	0.38
Observations	360	358	358	360	358	358
**B**	**Cognition**			**Depression**		
	**Model 7**	**Model 8**	**Model 9**	**Model 10**	**Model 11**	**Model 12**
Parents’ educational expectations	3.49 *** (2.73–4.24)			−3.00 *** (−3.72,−2.28)		
Children’s educational expectations		2.46 *** (1.70, 3.22)			−2.07 *** (−2.70, −1.44)	
Educational expectations gap			−2.95 ** (−5.24, −0.66)			1.85 ** (0.30, 3.40)
covariates	Yes	Yes	Yes	Yes	Yes	Yes
Constant	−1.60 (−2.68, 5.87)	3.19 (−1.17, 7.54)	11.76 (7.04, 16.49)	18.40 (14.50, 22.30)	16.47 (12.70, 20.25)	10.34 (6.62, 14.05)
R^2^	0.77	0.71	0.71	0.34	0.41	0.40
Observations	647	645	645	647	645	645

Note: Coefficients, 95% confidence intervals, and *p*-values are reported in this table. * *p* < 0.1. ** *p* < 0.05. *** *p* < 0.01. The model uses robust standard errors.

## Data Availability

The data are available online at http://www.isss.pku.edu.cn/cfps/ (registration and approval needed, accessed on 5 August 2022).

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
