# Peer review of "The Effect of Educational Expectations on Children’s Cognition and Depression"

_ijerph, 2022, doi:10.3390/ijerph192114070_

Round 1

Reviewer 1 Report

Dear Authors

Thanks for your valuable study.

Pleas mention the type of study ( e.g. Cohort, cross-sectional or analytical study) and your limitation, your including and excluding criteria clearly. 

Please update your references as 70% for  5 last years.

Please mention you hypothesis and proceed the part of discussion based on.  

Author Response

Dear professor:

Thank you very much for your valuable comments.Your comments have been very helpful in refining this research.We have carefully revised and responded carefully in accordance with your comments.At the same time, we marked the modified part in red in the original article.Please see the attachment for specific responses.Thank you again.

Reviewer 2 Report

The manuscript titled “The Effect of Educational Expectations on Children’s Cognition and Depression” aimed to investigate the effect of parents’ and children’s educational expectations on children’s cognition and depression. The study was based on data from two waves (2012 and 2016) of China Family Panel Studies (CFPS). Authors performed linear fixed-effects models to test their hypotheses, considering the following variables: a) cognition as dependent variable, b) depression as dependent variable, c) Parents’ educational expectations and children’s educational expectations as independent variables. In addition, they performed other analyses taking into account the difference between educational expectations pf parents and children (i.e., independent variable) and urban-rural differences (i.e., independent variable). Their findings showed that both parents’ and children’s educational expectation affected children’s cognition and depression. Additionally, differences between urban and rural environments on children’s cognition and depression were found. Authors discussed their results in light of previous literature as well as limitations and hints for future research.

I carefully read the manuscript and I think it may be of interest for readers of International Journal of Environmental Research and Public Health. Nevertheless, it could be worth considering some major points before it can be published as a research article. Below there are my comments and suggestions.

Introduction section

The Author(s) clearly presented the aims and the novelty of the study.

Lines 54-75, page 2: The Author(s) introduced the studies on parents’ educational expectations. They should consistently specify the role of parents’ educational expectation on children’s development, as well as the role of self-educational expectations.

Materials and Methods section

As a general indication and when applicable, please add proper citations throughout this section regarding tests and questionnaires employed.

Measures subsection

Lines 110-124, page 3: The Mini-Mental State Examination (MMSE) and CFPS data were combined in this study. What does it mean? In which way the final score was obtained?

Additionally: a) Could the Author(s) specify if standardized tools were included in the CFPS data? Were the MMSE score and CFPS data scores separately considered in the analyses? b) The total cognitive ability score is the sum of word recall score and the sequence score: what about the MMSE score? c) The MMSE is commonly used for adults or older adults. Which MMSE validation for children was used? Please, the Author(s) should report it; d) This section titled “Cognition”, but except the MMSE, which is a screening tools, only two cognitive measures were employed and not all cognitive domains were investigated.

Moreover, please specify the procedure for obtaining the final scores for each measures employed.

Lines 134-138, page 3: Who filled out this section of the CFPS questionnaire? Both parents? If yes, how the answers of both parents were considered in multi-categorical variable? Please, specify it.

Covariates subsection: there are some interesting covariates, even though come of them were never mentioned in the introduction section. Please specify in the Introduction section the theoretical basis for inserting such covariates and whether there are specific hypothesis about their impact on the relationship between independent and dependent variables.

2.2.5. Difference in educational expectations(independent variable)

Please provide an explanation for the calculation of the educational expectation difference. In particular, I do not understand the choice of reducing a continuous variable into a dichotomous one. If you can have a more precise (i.e., more than one point of difference) measure of the gap between the two expectations, why reducing such difference only to the values of 0 and 1?

Moreover, I think that the subsection “2.3 Methods” could be renamed into “2.3 Statistical Analysis”. Please, provide a more detailed specification about the plan of the statistical analysis. In particular, I do not fully understand the choice of performing two linear models, one for parents’ and the other for children’s expectations, and a third one with the gap between these two measures. Please, provide an explanation why it would be not sufficient to performs only the third model.

Results section

Lines 165-200, page 5 and 6: Results for Cognition were reported, but the Author(s) did not specify results for MMSE. Furthermore, no results for CFPS questionnaire on mental health were reported.

Moreover, it would be useful to report R-squared for each outcome in the tables or alternatively in the text.

Discussion

Lines 257-292, page 9: The Author(s) should better discuss these findings. They should explain the results and not just describe it. For example, parents’ educational expectations and children’s educational expectation had a negative effect on children depression. What does it mean? Which could be the mechanisms leading to these results? In which way parents’ and children’s expectations affect cognition and/or depression?

Conclusion

What could be the practical implications of this study?

Author Response

Dear Professor:

Thank you very much for your valuable comments.Your comments have been very helpful in refining this research.We have carefully revised and responded carefully in accordance with your comments.At the same time, we marked the modified part in red in the original article.Please see the attachment for specific responses.Thank you again.

Round 2

Reviewer 2 Report

I carefully read the revised version of the manuscript. I think that Authors have addressed all the issues raised by reviewers in first round of revision, and that the manuscript can be now published as a research article.